# The Value of Incorporating Social Preferences in Dynamic Ridesharing

**Sandhya Saisubramanian[1]    Connor Basich[1]    Shlomo Zilberstein[1]    Claudia V. Goldman[2]**

[1]College of Information and Computer Sciences, University of Massachusetts Amherst, USA

[2]General Motors, Advanced Technical Center, Israel

{saisubramanian, cbasich, shlomo}@cs.umass.edu, claudia.goldman@gm.com

## Abstract

Dynamic ridesharing services (DRS) play a major role in improving the efficiency of urban transportation. User satisfaction in dynamic ridesharing is determined by multiple factors such as travel time, cost, and social compatibility with co-passengers. Existing DRS optimize profit by maximizing the operational value for service providers or minimize travel time for users but they neglect the social experience of riders, which significantly influences the total value of the service to users. We propose DROPS, a dynamic ridesharing framework that factors the riders' *social preferences* in the matching process so as to improve the quality of the trips formed. Scheduling trips for users is a multi-objective optimization that aims to maximize the operational value for the service provider, while simultaneously maximizing the value of the trip for the users. The user value is estimated based on compatibility between co-passengers and the ride time. We then present a real-time matching algorithm for trip formation. Finally, we evaluate our approach empirically using real-world taxi trips data, and a population model including social preferences based on user surveys. The results demonstrate improvement in riders' social compatibility, without significantly affecting the vehicle miles for the service provider and travel time for users.

## Introduction

Dynamic ridesharing services, such as UberPool and Lyft-Line, are becoming an increasingly popular means of commute, especially in large cities (Chan and Shaheen 2012; Bathla et al. 2018). Dynamic ridesharing is characterized by matching multiple requests that arrive in real-time, for a one-way and one-time trip. We consider a setting in which a service provider operates a vehicle fleet and schedules cars to pick up and drop off passengers in response to a stream of requests, which includes matching requests with each other. There are two important factors that explain the growing attractiveness of DRS for customers: (i) cost effectiveness and (ii) ease of finding a ride in large cities where it is comparatively hard to find a taxi otherwise. For a service provider, dynamic ridesharing helps serve customers with possibly fewer vehicles, thus reducing their operational cost.

A common objective for optimizing riders' satisfaction in existing ridesharing systems is to minimize travel time (Ma, Zheng, and Wolfson 2013; Agatz et al. 2012; Bathla et al. 2018). In practice, however, there are many other factors that affect user satisfaction in dynamic ridesharing, apart from

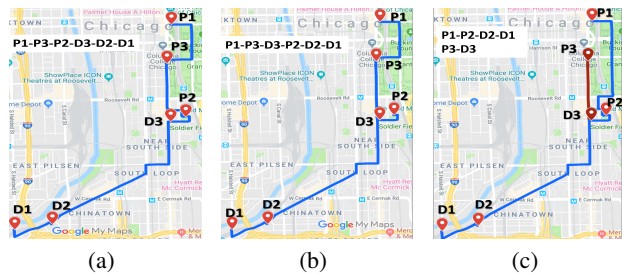

Figure 1: An example illustrating the influence of social preferences in trip formation. P denotes a pickup location and D denotes a dropoff location. A trajectory that maximizes operational value to the service provider is shown in (a). Incorporating and satisfying users' social preferences may lead to modification in the trajectory (b) or result in a different trip formation (c).

travel time. Since a user could be traveling with strangers in the ride, their compatibility plays a major role in the user's satisfaction. In fact, there is growing evidence that desire for personal space and security when riding with strangers pose a major barrier to using ridesharing for many users (Tao and Wu 2008; Agatz et al. 2012). For example, a female passenger may prefer to ride only with female co-passengers. The user may have a different set of preferences depending on the time of day and the location — preferences are trip-specific and not necessarily user-specific.

Consider a scenario with three requests where $r_1$ and $r_2$ are male and $r_3$ is a female passenger. Let these requests arrive at the same time (Figure 1), such that optimizing the operational value for the service provider forms a trip with these requests (1(a)). However, this may violate the users' social preferences and the trip may need to be altered to satisfy the preferences, such as the following:

- If the passengers prefer riding with co-passengers of the same gender but are indifferent to riding with co-passengers of a different gender, then it is desirable to minimize their ride time overlap in the vehicle by altering the pick up and drop off order (Figure 1(b)); and

- When the riders prefer co-passengers of the same gender and wish to *avoid* co-passengers of other gender, then it is better to form two trips (Figure 1(c)).

If the service does not provide a mechanism to express such social preferences and forms trips that violate these preferences (as in 1(a)), the customers may not use the service. Current DRS, however, do not account for social preferences in their optimization, despite being indicated as a major concern for users in several surveys (Agatz et al. 2012; Michalak et al. 1994; Furuhata et al. 2013; Svangren, Skov, and Kjeldskov 2018).

We present DROPS (**D**ynamic **R**idesharing **O**ptimization using social **P**reference**S**), a dynamic ridesharing framework that facilitates incorporating social preferences of the users in the trip formation process. A weight vector over preferences indicates the importance of each factor in determining the trip value to the user. The goal is to form trips that optimize both operational value for the service provider and value of the trip to the passengers, which incentivizes them to continue using the service and benefits the service provider. The value of a trip to a user is calculated based on their social compatibility with other co-passengers, the ride time, and ride cost. We solve this *bi-objective* optimization problem using scalarization (Roijers et al. 2013), which solves a linear combination of the multiple objectives. The relative importance of each objective can be controlled using the weight vector for the objectives. Given a set of riders, we evaluate their potential shared trip using an optimal trajectory planning algorithm. Candidate trips are formed using our real-time greedy algorithm that adds customers to a trip only if the trip's value is above a certain threshold.

We consider three basic social factors — age, gender, and user rating— along with a time preference indicating if the user is in a rush. The viability of factoring social preferences into the trips scheduling process is evaluated empirically. The experiments examine the impact of matching with social preferences (social matching) on users and the service provider. We test our approach on a *real-world taxi trips dataset* and compare the results with that of three baselines, each focusing on optimizing different components of the objective for trip formation. The population model and preferences used in our experiments were acquired using web-based *user surveys*, which was conducted in two phases and had 489 responses. The survey was conducted specifically to determine how different potential riders evaluate social ridesharing. Our results show that incorporating social preferences of users in the trip formation improves the overall user satisfaction, without significantly affecting the operational cost for the service provider.

Our primary contributions are: (i) presenting DROPS, a system for dynamic ridesharing with social preferences; (ii) proposing a real-time greedy algorithm for trip formation; and (iii) extensive empirical evaluation showing the benefits of social matching in dynamic ridesharing using real-world taxi data and a population model based on user surveys.

## Related Work

Dynamic ridesharing has gained popularity since the early 2000's due to the cost benefits it offers to the users and service providers, apart from its contributions to sustainable environment resulting from efficient vehicle usage. Dynamic ridesharing is characterized by user requests that ar-

rive in real-time and are matched with vehicles (Levofsky and Greenberg 2001). Another popular ridesharing setting is the car-pooling where users travel together for a particular purpose and the trips are usually recurring (Chan and Shaheen 2012). Our work differs from car-pooling as we focus on a dynamic ridesharing setting with a service provider who operates the vehicle fleet instead of individual car owners and trips that are typically non-recurring.

Optimizing dynamic ridesharing services has been an active research area, attracting researchers from diverse fields such as operations research, transportation, and artificial intelligence (Agatz et al. 2012; Chan and Shaheen 2012; Di Febbraro, Gattorna, and Sacco 2013; Alonso-Mora et al. 2017). Existing literature on dynamic ridesharing can be classified broadly based on the objective function and the solution method employed. Optimization-based approaches are the common solution technique employed (Santos and Xavier 2013; Ma, Zheng, and Wolfson 2013; Di Febbraro, Gattorna, and Sacco 2013; Biswas et al. 2017; Alonso-Mora et al. 2017; Dickerson et al. 2018; Bei and Zhang 2018). Other approaches include partition-based (Pelzer et al. 2015), auction-based mechanisms (Cheng, Nguyen, and Lau 2014), and genetic algorithms (Herbawi and Weber 2012). Researchers have employed these techniques largely to optimize the routing or travel time (Furuhata et al. 2013; Agatz et al. 2012; Herbawi and Weber 2012; Pelzer et al. 2015; Santos and Xavier 2013; Biswas et al. 2017). Specifically, the commonly used objectives for determining ridesharing matches are: (i) minimizing system-wide vehicle-miles; (ii) minimizing system-wide travel time; and (iii) maximizing number of participants.

A critical missing component of these objectives is the in-ride user experience. Numerous studies have outlined the need for learning and understanding user preferences in the context of ridesharing, beyond simple factors like time windows (Chan and Shaheen 2012; Agatz et al. 2012; Thaithatkul et al. 2015). Multiple surveys have acknowledged that it is essential to account for users' social preferences to improve dynamic ridesharing (Agatz et al. 2012; Di Febbraro, Gattorna, and Sacco 2013; Furuhata et al. 2013; Selker and Saphir 2010; Chan and Shaheen 2012; Montazery and Wilson 2016; Tao and Wu 2008; Miller and How 2017; Svangren, Skov, and Kjeldskov 2018; Bistaffa, Farinelli, and Ramchurn 2015). To address this discrepancy, we present a dynamic ridesharing framework that allows for representing and satisfying the social preferences of the users in trip formation.

## Problem Formulation

The DROPS framework facilitates customizing rides to improve user compatibility by incorporating the social preferences of users. Let $\mathcal{R}^t$ denote the finite set of unserved (non-dispatched) requests at time $t$ and $\mathcal{V}^t$ denote the finite set of available vehicles at time $t$. Each request $r \in \mathcal{R}^t$ is denoted by $\langle s, e, i, \vec{p}, U \rangle$. Each vehicle $v \in \mathcal{V}^t$ is denoted by the tuple $\langle ID, \omega \rangle$. Refer Table 1 for the definitions of variables and constants employed in the formulation.

We consider social preferences in each request that correspond to three social factors: age, gender, and rating of users.

Additionally, we consider a time preference to indicate if the user is in a rush. We identified these factors based on the results of our user surveys, conducted specifically to determine user expectations in ridesharing services. The preferences ($\vec{p}$) are denoted as $+1$, $-1$, or $0$, indicating the user's desirability, undesirability, or indifference about a certain value of a factor. For example, a preference of $+1$ for $rating \geq 4$ denotes that the person prefers riding with co-passengers who have a minimum rating of 4, and a preference of $-1$ for $rating \leq 3$ denotes that the person wishes to avoid riding with co-passengers who have a rating of 3 or below. That is, if rating on a scale of 1 to 5 is treated as a vector, then these preferences are denoted as $\langle -1, -1, -1, +1, +1 \rangle$. The weights $\vec{w} = [w_t, w_a, w_g, w_s]^T$ correspond to the time, age, gender, and rating, respectively.

A solution to an instance of this problem is a set of trips $\Lambda$, where each trip $\lambda \in \Lambda$ is a matching of requests to a vehicle and is denoted by $\langle R, v, \tau \rangle$. The value of a trip is denoted by $V(\lambda)$. The objective is to maximize the cumulative value of all trips dispatched in a given horizon $H$,

$$\max \sum_{t \in H} \sum_{\lambda \in \Lambda^t} V(\lambda).$$

**Multi-objective formulation**  Since the goal is to schedule trips that maximize the operational value for the service provider as well as maximizing the overall user value, this is naturally a bi-objective optimization. To solve this, we employ scalarization (Roijers et al. 2013), which projects a multi-objective value to a single scalar value by parameterizing the objectives using a weight vector. The weight value for each objective indicates its relative importance, thus resulting in a single objective function for optimization. Let $\beta_o$ denote the weight corresponding to the operational value and let $\beta_u$ denote the weight corresponding to the user value. Then, $\forall \lambda$, the trip value is:

$$V(\lambda) = \beta_o \underbrace{\sum_{r \in R_\lambda} (x_r - d_r) - c_\lambda^\tau}_{\text{operational value}} + \beta_u \underbrace{\sum_{r \in R_\lambda} (\alpha_r + d_r)}_{\text{user value}} \quad (1)$$

The operational value and the user value are measured in dollars (\$) and normalized to the same scale before scalarization. The operational value to the service provider depends on the cost of operating the vehicle for the trip $c_\lambda^\tau$ and the amount paid by the riders, which is the difference between the amount charged for the trip ($x_r$) and the discount offered for using ridesharing ($d_r$). The value of the trip to a user depends on the user utility ($\alpha_r$) and the discount gained for using ridesharing ($d_r$). The user utility ($\alpha_r$) is the difference between the users' social compatibility with all their co-passengers and the extra travel time incurred by using ridesharing. The social compatibility for a request is calculated as the cumulative weighted difference between the preferences $\vec{p_r}$ and demographics of each co-passenger.

We now explain the social utility calculation using a simple example. Consider two requests $r_1$ (female) and $r_2$ (male) that arrive at the same time and have the same source and destination coordinates, same age (30), and rating (4).

| Variables | Definitions |
|---|---|
| $\Lambda^t$ | Set of trips formed at time $t$ |
| $V(\lambda)$ | Value of trip $\lambda$ |
| $\beta_o, \beta_u$ | scalarization weights |
| $R_\lambda = \{r_1, \ldots r_k\}$ | Set of requests matched for the trip |
| $c_\lambda^\tau$ | Cost of using the vehicle for the trip corresponding to the ride route $\tau$ |
| $\omega_v$ | Passenger capacity of vehicle |
| $s_r, e_r$ | Start (pick-up), end (drop-off) locations of $r$ for the trip |
| $\alpha_r$ | User's social utility |
| $x_r$ | Amount charged to $r$ for the trip |
| $d_r$ | Discount for using ridesharing |
| $i_r$ | Request initiation time |
| $\vec{p_r}$ | Social and time preferences of $r$ |
| $\vec{w_r}$ | User's weights for preferences $\vec{p_r}$ |
| $U_r$ | User demographics: {age, gender, rating} |
| $ID_v$ | Vehicle ID |

Table 1: Notations

$r_1$ prefers ($+1$) female co-passengers with age in the range 20-40 with rating $\geq 4$ and expresses undesirability ($-1$) for all other values of social factors. Let the weights of these social preferences be $\vec{w_{r_1}} = [0.3, 0.3, 0.2, 0.5]^T$, corresponding to time, age, gender, and rating. The social compatibility for $r_1$ with respect to $r_2$ is $0.3 - 0.2 + 0.5 = 0.6$. Let the extra trip time be 2 minutes, then $\alpha_{r_1} = 0.6 - 0.3 * 2 = 0$.

## Solution Approach

Given a set of requests and vehicles, our solution approach consists of two components: (i) trip formation and (ii) trip dispatch. Figure 2 is an illustration of our solution approach. In each decision cycle, the trip formation component matches requests with each other and to vehicles, and the dispatch component decides which trips are dispatched. We restrict the scope of matching in this paper to requests and vehicles that have not been dispatched. That is, we do not consider a vehicle *en-route* (already driving on the road) in the scheduling process and therefore do not match requests to such vehicles. The route planner calculates the optimal trajectory for picking up and dropping off a given set of requests.

### Trip Formation

In this phase, requests are matched with other requests and assigned a vehicle to form a trip. The matching is performed using a greedy approach outlined in Algorithm 1. The input to the algorithm is the set of requests and a trip value threshold $\delta$ that indicates the required minimum improvement in trip value to form trips. The algorithm adds a request to the best trip (maximum improvement) that improves the trip value at least by a factor of $\delta$ and if the trip size has not exceeded the maximum capacity of the vehicle (Lines 7-16). Standard hyperparameter tuning or sample average

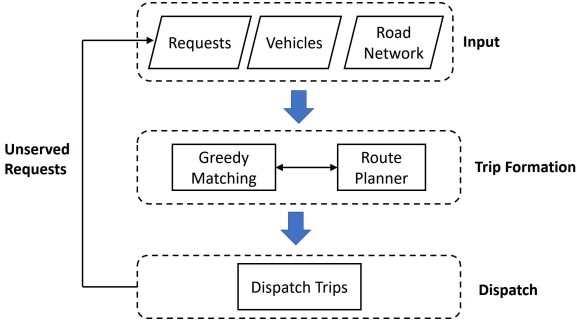

Figure 2: Overview of the solution method.

---

**Algorithm 1: Greedy Matching** $(\mathcal{R}^t, \delta)$

1   $\Lambda^t \leftarrow \emptyset$
2   **foreach** $r \in \mathcal{R}^t$ **do**
3      matched = false
4      **if** $|\Lambda^t| > 0$ **then**
5         $\lambda_{best}, \lambda_{rem} \leftarrow \emptyset$
6         Best_Value $= -\infty$
7         **foreach** $\lambda \in \Lambda^t$ *with* $|R_\lambda| < \omega_\lambda$ **do**
8            Calculate best route for $\lambda' = \lambda + r$
9            **if** $\frac{V(\lambda') - V(\lambda)}{V(\lambda)} \geq \delta$ *and* $V(\lambda') > $ *Best_Value*
            **then**
10               $\lambda_{rem} \leftarrow \lambda; \lambda_{best} \leftarrow \lambda'$
11               Best_Value $= V(\lambda_{best})$
12               matched = true
13            **end**
14         **end**
15         **if** *matched = true* **then**
16            $\Lambda^t \leftarrow (\Lambda^t \backslash \lambda_{rem}) \cup \lambda_{best}$
17         **end**
18      **end**
19      **if** *matched = false* **then**
20         Create new trip $\lambda$ with request $r$
21         Calculate best route for $\lambda$
22         $\Lambda^t \leftarrow \Lambda^t \cup \lambda$
23      **end**
24   **end**
25   **return** $\Lambda^t$

---

approximation (Kleywegt, Shapiro, and Homem-de Mello 2002) may be used to estimate $\delta$. The trip value is estimated using Equation 1.

Each request is assigned to the best trip that satisfies the threshold improvement. If no such trip is found, then a new trip is created with the request (Lines 19-22). This ensures that all requests are associated with a trip. The route planner computes trajectories that determine the pick up and drop off order for a given set of requests. All possible trajectories are generated and the one that maximizes the trip value for a given set of requests is selected as the route $\tau$ for the trip. During the trip formation, the best route is updated whenever a new request is added to a trip (Line 8, 21). The output of this algorithm is the set of all trips formed, $\Lambda^t$.

**Partitioning Requests for Scalability**   The computational complexity of the matching algorithm discussed above increases rapidly with the increase in number of requests. To counter this computational complexity, we exploit the notion of independence among requests. Two requests $q$ and $r$ are said to be independent if serving them together in the same trip is not desirable in terms of trip value. Hence, all the requests over different days or requests with non-overlapping source-destination pairs are independent. The requests can be partitioned based on their dependence and matches may be formed among each partition in parallel. Sometimes, it is non-trivial to estimate an exact partitioning of requests with respect to routes, without forming trips and calculating the best route possible. In such cases, the underlying map may be partitioned into geographic zones to form trips in each zone independently by considering the requests originating in that zone, as in our experiments.

## Trip Dispatch

The trips formed in the matching phase are dispatched in this phase if at least one of the following conditions is satisfied: (i) trip value is above the predefined dispatch threshold; or (ii) a request in the trip has remained unserved for a certain period of time since its arrival (queue time). The dispatch threshold for trip value and the queue time for the requests are determined by the service provider. For example, all requests that are unserved for five minutes or more since their arrival time may be dispatched irrespective of the trip value, depending on vehicle availability. In our experiments, trips

that satisfy the queue time threshold are given a higher priority over the trips with lower queue time but higher trip value. This ensures that certain requests do not remain unserved forever due to lower trip value. The trips are then dispatched based on availability of vehicles, $\mathcal{V}^t$. At the end of decision cycle $t$, all unserved requests — requests in trips that are not dispatched — are added to the requests set for the next decision cycle, $\mathcal{R}^{t+1}$.

## Experimental Results

The experiments evaluate the impact of using social preferences in ridesharing, with respect to users and the service provider. We built a realistic simulator of ridesharing using the Chicago taxi trips dataset[1] and a population model based on extensive user surveys. We compare the results obtained using social preferences in dynamic ridesharing matching (SM) with that of three baselines: $(B_1)$ maximizing only the operational value, $\beta_u = 0, \beta_o = 1$; $(B_2)$ maximizing only user value, $\beta_u = 1, \beta_o = 0$; and $(B_3)$ maximizing the comprehensive trip value in Equation 1 but without considering user's social preferences corresponding to age, gender, and rating, $w_a = 0, w_g = 0, w_s = 0$, for the trip formation. Algorithm 1 is used to form trips using each baseline objective.

The algorithms and the simulation system were implemented by us on an Intel Xeon 3.10 GHz computer with 16GB of RAM, using a homogeneous vehicle fleet with a

---

[1]https://data.cityofchicago.org/Transportation/Taxi-Trips/wrvz-psew

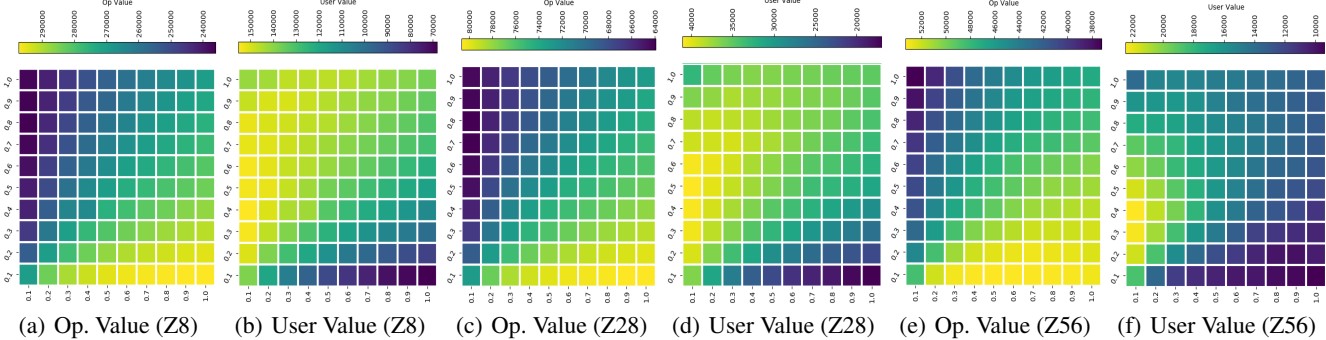

| (a) Op. Value (Z8) | (b) User Value (Z8) | (c) Op. Value (Z28) | (d) User Value (Z28) | (e) Op. Value (Z56) | (f) User Value (Z56) |

Figure 3: Heat map of the operational (Op.) value and user value corresponding to different weights in each zone.

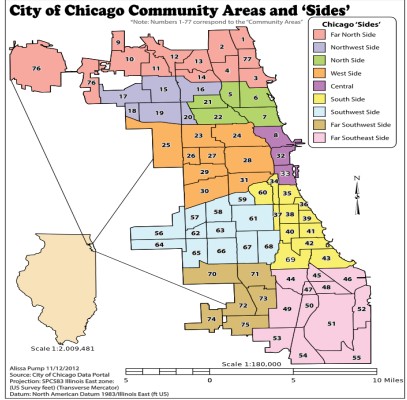

Figure 4: A map of Chicago divided into zones.

seat capacity of 4 for the evaluation. Each decision cycle is 30 seconds in real-time and the horizon $H$ is one day. We assume that the number of vehicles is not bounded since the benefits of social matching are best illustrated in this case and all techniques are equally affected by vehicle restriction. We set the trip threshold $\delta$ to zero for the greedy algorithm; requests are added to the best trips possible as long as the current value of the trip is not diminished. This allows us to examine the benefit of social matching uniformly across zones by using a conservative value. However, in practice this hyperparameter may be tuned to further optimize performance subject to the service provider's objective. The request queue time threshold for dispatch is set to five minutes. The travel time and distances are calculated using straight line distances between the coordinates and a vehicle speed of 30 miles per hour. While these experiments do not account for the actual routes and traffic conditions, these factors are not likely to change the relative merits of each approach and the conclusions of the study.

### Population Model and Dataset

The population model considered in our experiments is based on the results of online surveys that was conducted in North America. The survey had 489 responses which indicated that users would like to be matched with people who are similar to them. The demographic information such as age and gender, for our experiments, is drawn from the ac-

tual Chicago demographic distributions[2]. The preferences ($\vec{p}$) and the weights ($\vec{w}$) are based on the survey results. The survey also indicates that some users are unwilling to use ridesharing when social preferences are not taken into account. To reflect this, certain users were marked as reluctant for ridesharing in the absence of social matching and these users were always dispatched as solo rides, when forming trips with the baseline objectives.

The Chicago taxi trips data consists of trip-specific information such as start time and end time of the taxi ride, trip duration, trip fare, and the latitude and longitude coordinates for pick up and drop off locations along with the geographic zone corresponding to these locations. A map of Chicago divided into zones[3] is shown in Figure 4. We partition the data from each zone into training and testing sets. The weights for scalarization were estimated empirically using the training data (Figure 3). In Figure 3, the x-axis is the weight for operational value ($\beta_o$) and the y-axis denotes the weight corresponding to user value ($\beta_u$). The weights used for the test sets are $\beta_o = 0.8$ and $\beta_u = 0.6$ for zones 8 and 28, and we used $\beta_o = 0.5$ and $\beta_u = 0.5$ for experiments on zone 56. Our algorithm is evaluated along different metrics on the test set which uses data from two consecutive weeks in April 2015. We consider requests originating in zones 8, 28, and 56, whose requests densities are high, medium, and low respectively. The average number of requests per day in each of these zones is 20000, 7000, and 1500 respectively.

### Analysis of Tradeoffs

Since user value and the operational value are often competing metrics, we analyze the quality of trips formed with respect to each of these.

**Impact on Users** We measure the impact on users based on the total user value (Figure 5), average social utility per minute (Figure 6), and the increase in ride time, relative to a solo trip (Figure 7).

Trips formed by maximizing operational value ($B_1$) have the least user value across all zones, as expected. Our approach (SM) achieves user value close to that of optimiz-

---

[2]http://chicago.areaconnect.com/statistics.htm

[3]https://en.wikipedia.org/wiki/Community_areas_in_Chicago

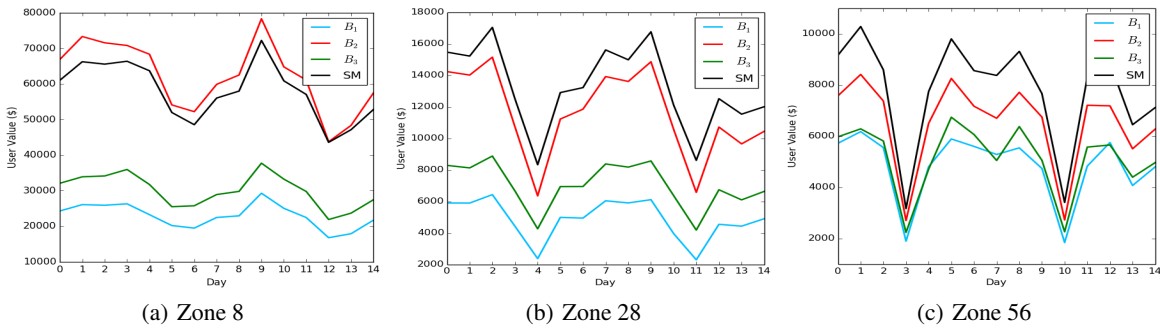

(a) Zone 8        (b) Zone 28        (c) Zone 56

Figure 5: Total user value of trips dispatched on each day.

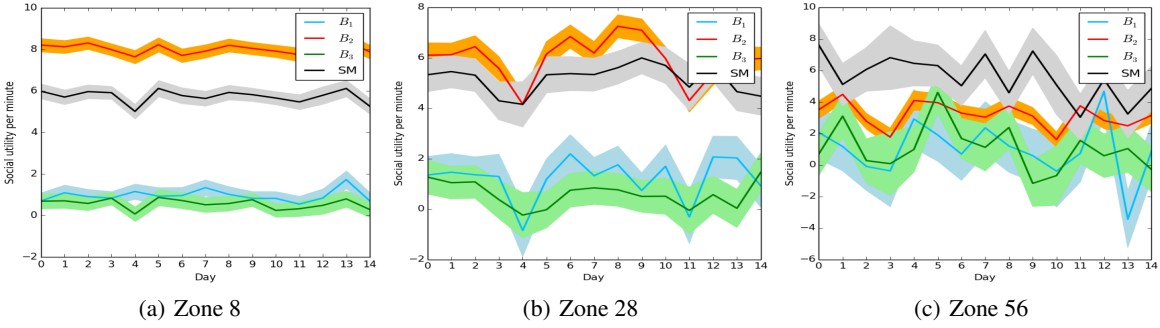

(a) Zone 8        (b) Zone 28        (c) Zone 56

Figure 6: Average social utility per minute for users on each day. This measures the social compatibility of users with their co-passengers in the trips formed using each objective function.

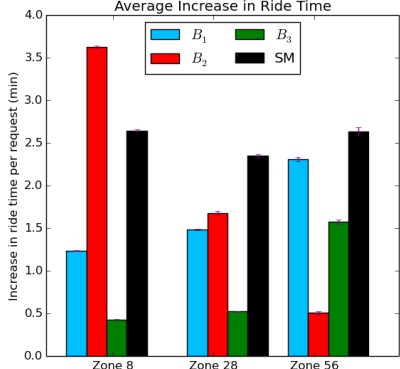

Figure 7: Average increase in ride time of users compared to solo rides. Our approach, on average, increases the ride time by at most 2.5 minutes, well within the acceptable range found in our user surveys.

ing for user value alone ($B_2$), and sometimes better than $B_2$. This is because, in some cases, the values of the trips formed by optimizing $B_2$ objective may not meet the dispatch threshold in which the case the trips are dispatched after five minutes, which eventually reduces the user value. Our approach overcomes this drawback by optimizing for both the objectives, providing greater cumulative value for a given trip and enabling it to be dispatched more quickly.

The social utility ($\alpha_r$) per minute measures the average social compatibility of users with their co-passengers. To account for the different ride times of the trips, we measure the average utility per minute, along with standard error (Figure 6). We observe that SM consistently performs similar to or better than $B_2$, showing that the user value is improved through better matching, and not merely based on the ride time or discount offered.

We also evaluated the increase in ride time of the different techniques, compared to solo ride (Figure 7). The average ride times are in the range 10-20 minutes for requests originating in zones of interest. Though the increase in ride time of our technique is around three minutes, note that ridesharing, in general, incurs additional ride time. The increase in ride time of our technique is well within the range that users consider acceptable (at most 5 minutes) according to the survey results. The social compatibility typically offsets the increase in ride time for the users, thus resulting in increased user utility when forming trips using our approach.

**Impact on the Service Provider** The impact on service provider is determined based on the operational value and the total miles driven, to give a sense of degree of variation induced by social matching on the trip routes and quality of service. As expected, objective $B_1$ achieves the highest operational value and maximizing $B_2$ objective has the lowest operational value (Figure 8). The operational value achieved by our approach (SM) is closer to that of $B_1$, with a slightly higher miles driven (Figure 9) and higher user utility. The total number of trips formed by our approach is also comparable to that of $B_1$. This shows that our approach improves the quality of trips without significantly affecting the total miles driven or the cost of operating the service by the provider.

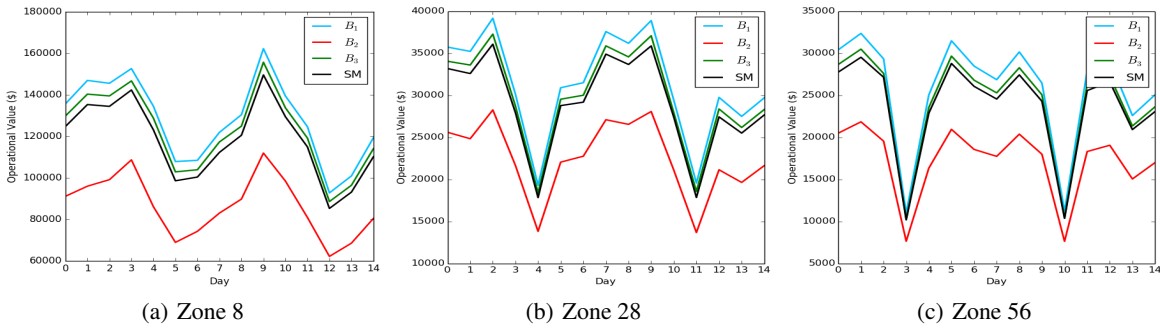

|  |  |  |
|:---:|:---:|:---:|
| (a) Zone 8 | (b) Zone 28 | (c) Zone 56 |

Figure 8: Total operational value of trips dispatched on each day.

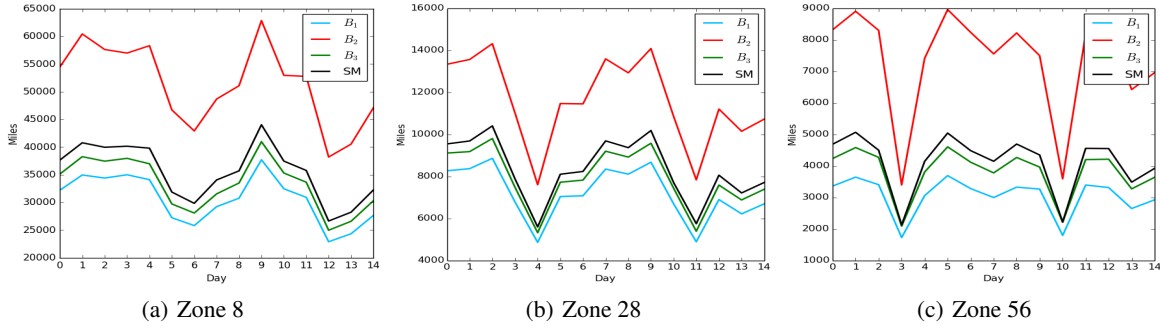

|  |  |  |
|:---:|:---:|:---:|
| (a) Zone 8 | (b) Zone 28 | (c) Zone 56 |

Figure 9: Total miles driven on each day.

## Scalability and Robustness

Since matching is performed every 30 seconds, it is important to ensure that the matching algorithm is fast so that it may be effectively used in real-time. The run time (in seconds) of our matching algorithm is $0.5$ on average in the zone with high request density (zone 8), $0.12$ in zone 28, and $0.003$ in zone 56, demonstrating the scalability of DROPS.

We also compared our matching algorithm to a hindsight greedy matching with access to all the requests in a day, including future ones. The purpose of this experiment is to evaluate the gain in operational value and user value that could be achieved when knowledge of future requests is available. We compare the total operational value obtained using our approach with that of optimizing only for operational value with all requests in a day. Similarly, the total user value obtained with our approach, with requests arriving in real-time, is compared with that of optimizing for user value only and with access to all requests in a day. Trips are formed using the best-fit greedy algorithm (Algorithm 1) for our approach and for the hindsight evaluation.

The results, summarized in Table 2, show that our approach achieves at least ∼89% of the operational value and up to ∼84% of the user value compared to the hindsight

| Zone
Metrics | Zone 8 | Zone 28 | Zone 56 |
|:---|:---:|:---:|:---:|
| Operational Value | 91.96% | 93.03% | 89.87% |
| User Value | 83.62% | 82.43% | 66.71% |

Table 2: Performance relative to hindsight optimization.

matching in all three zones, indicating that any prediction method of future requests would yield very limited performance gains in the operational value. However, some improvements in user value could be achieved with knowledge of future requests by forming trips where the users have a higher social compatibility with co-passengers.

## Conclusion and Future Work

Dynamic ridesharing is an increasingly appealing commuter option. However, numerous surveys have indicated that users' concerns, primarily about the social characteristics of co-passengers, pose a major barrier to using ridesharing for a segment of the population. We present the DROPS system for optimizing dynamic ridesharing with social preferences and present an efficient real-time matching algorithm that can handle effectively high density zones.

Our results demonstrate that factoring social preferences into the matching process helps improve the user value, without significantly affecting the operational value to the service provider. Furthermore, survey results indicate that services that perform social matching are likely to incentivize more individuals to use the service. We conclude that while social matching is beneficial overall, it is not always guaranteed to result in improved performance. Factoring social preferences into the matching process is most beneficial in zones with a high request density per decision cycle and greater compatibility among ridesharing users.

In the future, we aim to examine ways to extend the matching model to consider nearby trips that have already been dispatched and are currently en-route. We will also

consider more complex ways to factor the competing objectives using more general multi-objective planning algorithms (Wray, Zilberstein, and Mouaddib 2015). Additionally, based on the performance analysis of our approach with that of a hindsight trip formation, we aim to employ a predictive model for future requests to improve the user value. While we anticipate some performance gains, we do not expect the relative benefits of social matching to diminish.

## Acknowledgments

We thank Shannon Roberts and Fangda Zhang for conducting the user survey and for fruitful discussions on designing the population model.

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
