# OpenReview forum: "The Value of Incorporating Social Preferences in Dynamic Ridesharing"
_icaps-conference.org/ICAPS/2019/Workshop/SPARK — SPARK 2019_

### Official Review · AnonReviewer1 · 2019-05-01
**interesting work on a timely problem**

**Rating:** 5
**Confidence:** 2

**Review:**

This paper addresses the problem of optimizing dynamic ridesharing in a setting that accounts for social preferences of requestors.  The primary technical contribution is a heuristic, real-time algorithm for matching requests to drivers that seeks to balance operational utility (vehicle operator’s perspective) with user value.  The model of user value was derived from an extensive survey. A series of evaluations based on real-world taxi data shows that the approach improves user value without significantly affecting operational value for the service provider.

This is quite a nice paper:  interesting and timely topic area, technical solution well matched to the problem, solid experimental methodology, and good results.  The presentation and level of scholarship in the paper is high.

A couple of comments:
-	The paper states that 5 minutes was set as an acceptable threshold for increased ride time by users.  But surely that threshold would depend on the length of the ride?
-	The increase in ride time is simplistic in that it accounts only for distance traveled.  More realistically, there would also be “stopping time” associated with pulling over, passengers entering/exiting the vehicle, retrieving luggage from the trunk, etc.
-	The population model was derived from a large-scale survey with “demographic information ... drawn from the actual Chicago demographic distributions”.  Won’t those demographics differ from the demographics of people who use ride-sharing services?
-	The overall algorithm design separates Trip Formation from Trip Dispatch, leading to two separate heuristic planning algorithms.  Did you consider a uniform optimization algorithm that would combine both functions?

---

### Official Review · AnonReviewer2 · 2019-05-06
**Comments**

**Rating:** 4
**Confidence:** 2

**Review:**

Quality and originality:
The paper contributes well formulated study toward incorporating social preferences of users in ride sharing scheduling. It's a significantly untapped area of research particularly concrete algorithms and user studies wise. In this sense, authors did a nice job establishing state-of-the art results.

clarity:
The paper motivates the need for accounting user's social preference during ride scheduling really well. It is also formulate the problem precisely and experimentation and conclusion sections are easy to follow.

Significance:
The work in this paper is a crucial step towards improving ride sharing experience for users with minimal decrease in operation value for service provide and noticeable gains in social utility for users. Couple of areas that require further exploration is in terms of increase in ride time w.r.t. to average ride time.

Questions for Authors:
1. The example showing social compatibility calculation for r1 w.r.t to r2 is unclear given their request types. Can authors please elaborate it further? As part of the same example, "w_r1" value computation is unclear as well.

2. The Experimental section states that the number of vehicles is not bounded. Vehicle availability plays a crucial role in scheduling which in turn impacts operation margins. Can authors elaborate why were the number of vehicles left unbounded? or what will be implications of bounding them especially for greedy algorithm.

3. Reluctant users were always dispatched as solo riders in baseline models. Can authors give details about what percentage of these reluctant users were paired up in SM approach? Also what's the percentage of reluctant users to regular users?

4. Authors have highlighted that avg increased ride time was less than 2.5 mins but can they provided what was average ride time?

5. SM approach shows noticeably week performance in low density (zone 56) settings. User value and operation value w.r.t to optimal solutions is significantly lower along with identifiable difference in ride time w.r.t. to B3. Does authors have any explanation for this behavior?

---

### Public Comment · ~Christophe_GUETTIER1 · 2019-05-01
**General Comment**

An interesting paper with many insights on the problem of ride sharing among several co-passengers. The problem is well formulated and the state of the art is clear and well presented. Authors explore a model for RideSharing  confronting both operational and user values. They also propose a greedy algorithm and discuss its properties (robustness, saclability) as well as performances.

This paper also raises the question of social interactions and their corresponding mathematical models, discussed as social compatibility between co-passengers in the paper. However, the social interaction modelling level looks very simple and finally lost in a global cost function ... The concept should be of interest for discussions.

---

### Decision · Program_Chairs · 2019-05-08
**Acceptance Decision**

**Decision:**

Accept

**Comment:**

A welcome contribution to the workshop.